# Current Treatments and Innovations in Diabetic Retinopathy and Diabetic Macular Edema

**DOI:** 10.3390/pharmaceutics15010122

**Published:** 2022-12-29

**Authors:** Jesus H. Gonzalez-Cortes, Victor A. Martinez-Pacheco, Jesus E. Gonzalez-Cantu, Alper Bilgic, Francesc March de Ribot, Aditya Sudhalkar, Jesus Mohamed-Hamsho, Laurent Kodjikian, Thibaud Mathis

**Affiliations:** 1Ophthalmology Department, School of Medicine, University Hospital “Dr. Jose Eleuterio Gonzalez”, Universidad Autónoma de Nuevo León, Monterrey 64460, Mexico; 2Retina and Vitreous Department, Hospital de Nuestra Señora de la Luz, Universidad Nacional Autónoma de México, Mexico City 06030, Mexico; 3Ophthalmology Department, Instituto Avalos, University Galileo, Guatemala City 01010, Guatemala; 4Alphavision Augenarztpraxis, 27568 Bremerhaven, Germany; 5Department of Ophthalmology, Otago University, Dunedin 9016, New Zealand; 6Department of Ophthalmology, Girona University, 17004 Girona, Spain; 7MS Sudhalkar Medical Research Foundation, Baroda 390001, India; 8Service d’Ophtalmologie, Centre Hospitalier Universitaire de la Croix-Rousse, Hospices Civils de Lyon, Université Claude Bernard Lyon 1, 69004 Lyon, France; 9Unité Mixte de Recherche—Centre National de la Recherche Scientifique 5510, Matéis, Villeurbanne, 69004 Lyon, France

**Keywords:** anti-vascular endothelial growth factor, corticosteroids, dexamethasone-implant, diabetic macular edema, intravitreal injections, port delivery system, proliferative diabetic retinopathy

## Abstract

Diabetic retinopathy (DR) is one of the leading causes of blindness worldwide. Multiple treatment options have been used over time to attempt to modify the natural progression of the disease in both proliferative diabetic retinopathy (PDR) and diabetic macular edema (DME). These two retinal complications are the result of microvascular occlusions and vascular hyperpermeability and are considered one of the leading causes of irreversible blindness in patients of working age. It is now well demonstrated that PDR and DME are associated with increased levels of inflammatory and pro-angiogenic factors in the ocular compartment. To date, laser photocoagulation, vascular endothelial growth factor (VEGF) inhibitors, and corticosteroids have demonstrated efficacy in their treatment in large randomized controlled trials and in real-life observational studies. This manuscript aims to provide a comprehensive review of current treatments, including the main drugs used in diabetic pathologic manifestations, as well as new therapeutic alternatives, such as extended-release intraocular devices.

## 1. Introduction

Diabetes mellitus (DM) affects over 422 million people worldwide and almost 1.6 million deaths are directly related to the disease each year. The number of diabetic people is expected to increase to 700 million by 2045 [1]. Diabetic retinopathy (DR) is among the leading causes of acquired vision loss between the ages of 20 and 64 years in developed countries [2]. In non-proliferative DR (NPDR), diabetic macular edema (DME) is the major cause of vision loss [3] and is caused by hyperglycemia-induced damage to endothelial pericytes and tight junctions, which causes a dysfunctional inner blood-retinal barrier (BRB). Associated risk factors include elevated blood glucose, lipids and triglyceride levels, hypertension, advanced diabetic nephropathy, and pregnancy [4]. Excess glucose accumulation leads to the activation of two critical pathways in the pathophysiology of DR and DME: angiogenesis and inflammation [5,6,7].

Vascular endothelial growth factor (VEGF) and placental growth factor (PlGF) induce vascular hyperpermeability and microvascular changes; the former also stimulates angiogenesis and recruitment of inflammatory cells [7,8]. Hyperglycaemia also results in the formation of advanced glycation end products (AGEs) and reactive oxygen species (ROS), which lead to nitric oxide synthase dysregulation. This, in turn, activates pro-inflammatory transcription factors such as nuclear factor κappa-Beta (NFκ-β) followed by an increase in cytokines (IL-1, IL-6, TNFα), and chemokines, such as monocyte chemo-attractant protein 1 (MCP1), intercellular adhesion molecule-1 (ICAM-1) and vascular cell adhesion molecule-1 (VCAM-1). Furthermore, this activates endothelial cells, recruitment of inflammatory cells, and increases the level of VEGF. Finally, NFκ-β activation leads also to the synthesis of many proinflammatory molecules by promoting specific gene regulation [8].

Laser treatment for DME was established as the standard of care for nearly 30 years when the Early Treatment of Diabetic Retinopathy Study (ETDRS) was published in 1985 [9]. This study demonstrated that focal photocoagulation of “clinically significant” DME reduces the risk of visual loss, increasing the chances of visual improvement. Moreover, it also demonstrated decreased persistence of edema and less visual field loss. Panretinal photocoagulation (PRP) has been the standard treatment for PDR since the DRS study demonstrated its benefit more than 40 years ago [10]. However, PRP has demonstrated permanent peripheral visual field loss and decreased night vision. On the other hand, it could exacerbate existing DME or increase its incidence.

In 2005, the first pharmacological anti-angiogenic (anti-VEGF) drug was used for the treatment of DME. Anti-VEGF therapy in DME has shown superior visual acuity results and acceptable risks compared to focal or macular grid laser, and has also led to the observation that PDR lesions can be reversed during treatment [11,12,13,14]. Since the advent of anti-VEGF therapy, new alternatives have been developed in search of greater durability and efficacy [15,16]. However, patients need to be followed regularly for retinal assessment, retreatment, or monitoring of side effects. It was therefore clear that a more successful treatment for DME and PDR was needed, and different treatment alternatives for both diabetic retinal complications should be considered [17].

The purpose of this review is to provide the reader with a ready update in terms of currently available treatment modalities and current preferred practice patterns. The paper discusses, moreover, new avenues in diabetic retinopathy treatments and presents novel and future therapeutic options, as well as the area of research. Anti-VEGF molecules and steroid treatment were first described, as they are the current option available for the treatment of diabetic patients. Then we developed future therapies exploring new pathophysiologic areas.

## 2. Methods

This article is based on a review of the literature performed by the authors. A literature search was performed in July 2022 using PubMed to identify relevant publications related to DR and DME using the following search terms: “diabetic retinopathy” or “diabetic macular edema” associated with “treatment” or “anti-VEGF” or “steroids”. The aim of this review is to describe the different treatment options for DR and DME and to discuss specific management based on disease characteristics and patient profile.

## 3. Treatment Options

### 3.1. Anti-Vascular Endothelial Growth Factor

The main goal of anti-VEGF-based therapies is to block the activity of elevated concentrations of VEGF and restore BRB integrity. The first intravitreal drug used to treat DME was the pegylated anti-VEGF aptamer pegaptanib sodium (Macugen; Pfizer), which selectively blocks VEGF isoform 165 [12]. Its successors superseded the results obtained by blocking all VEGF isoforms (Table 1).

#### 3.1.1. Bevacizumab

Bevacizumab (Avastin; Genentech, Inc., South San Francisco, CA, USA) is a 149 kDa recombinant immunoglobulin G1 humanized monoclonal antibody that was originally approved by the U.S. Food and Drug Administration for the treatment of colorectal cancer in 2004, and its off-label intraocular use at doses of 1.25 to 2.50 mg in 0.05 mL had demonstrated good visual results and efficacy in the treatment of DME [13,14,27]. Bevacizumab binds and neutralizes the three biologically active isoforms of VEGF-A: VEGF_165_, VEGF_121,_ and VEGF_110_; the vitreous half-life is 9.8 days with a mean (SD) serum half-life of 18.7 (5.8) days in non-vitrectomized patients [28,29,30]. Bevacizumab proved to be superior to laser therapy which was considered the gold standard of treatment. In the 2-year results of the BOLT study, the bevacizumab arm showed significantly better visual gain than laser (which showed visual loss) [31]. This is unclear in the literature how bevacizumab is non-inferior to other anti-VEGF therapy to treat DME. In the 5-year extension of the randomized clinical controlled trial, DRCRNet protocol T, a significant difference compared to aflibercept and ranibizumab was found in the first 2 years but not in the 5-year extension of the protocol. There was no difference found between anti-VEGF in patients with BCVA 20/40 or better throughout the 2 years of the study. However, in eyes with BCVA of 20/50 or worse, aflibercept was superior to ranibizumab and bevacizumab at one year, whereas at 2 years aflibercept was no longer superior to ranibizumab, but remained superior to bevacizumab [20,32].

The recent DRCRNet protocol AC demonstrated the effectiveness of the initial treatment of bevacizumab with a switch to aflibercept if specific criteria were met, compared to aflibercept monotherapy for DME treatment at 2 years. Half the patients in each group had a ≥2-step improvement from baseline on the ETDRS-DRSS. However, it should be noted that approximately 70% of the patients in the bevacizumab-first switched to aflibercept during the 2-year trial [27]. These data demonstrate that even if it is not FDA-approved, bevacizumab keeps being a safe, effective, and cost-effective option for DME and DR management, at least at the beginning of the disease.

#### 3.1.2. Ranibizumab

Ranibizumab (Lucentis; Genentech, South San Francisco, CA, USA) is a 48 kDa anti-VEGF-A affinity-matured monovalent monoclonal antibody fragment designed for ocular use. The estimated vitreous half-life of ranibizumab is ~9 days, and due to the absence of the Fc antibody region, it is cleared from the bloodstream more rapidly and has a short systemic elimination half-life of ~2 h [28,33]. In 2012, it was the first FDA-approved anti-VEGF protein for the treatment of DME. The approval of ranibizumab for DME was based on three results of phase 3 clinical trials (RISE and RIDE) where 2 doses (0.5 or 0.3 mg) were compared to sham injections. The results showed that at 2 years, 44.8% and 45.7% of patients with monthly ranibizumab gained ≥15 letters with the 0.3 and 0.5 mg dosage, respectively, compared with 18.1% and 12.3% in the sham group. Furthermore, structural improvement in optical coherence tomography (OCT) was higher in all ranibizumab groups compared to sham in each measurement posttreatment. Resolution of leakage on fluorescein angiography (FA) and DME on OCT both were statistically significantly more common among ranibizumab-treated patients. Patients randomized to ranibizumab were less likely to develop proliferative diabetic retinopathy (PDR), had higher rates of retinopathy improvement, and lower rates of retinopathy progression [34]. As per DRCRNet protocol S, ranibizumab was shown to be non-inferior to PRP regarding BCVA outcome and visual field changes in patients with PDR. At 2 years, BCVA improved from baseline in the ranibizumab arm, and only stability was obtained from baseline in the PRP group. There was significant visual field loss in the PRP group, and more vitrectomies were required in the PRP group [32,35].

In October 2021, the FDA approved the port delivery system (PDS) ocular implant of ranibizumab (100 mg/mL) for the treatment of neovascular age-related macular degeneration (nAMD) (Susvimo; Genentech, South San Francisco, CA, USA). Currently, two phase 3 clinical trials are being carried out; PAGODA study is comparing Susvimo with a refill every 6 months in DME patients against monthly ranibizumab 0.5 mg, and PAVILION is evaluating the number of patients with ≥2-step improvement from baseline on the ETDRS-DRSS at one year with Susvimo in moderately severe or severe NPDR without DME [36,37]. However, due to concerns about the hermeticity of the seal that prevents the drug from leaking out after injection, in October 2022 the company pulled the product from the market and is expected to solve this problem within a year [38].

#### 3.1.3. Aflibercept

Aflibercept (Eylea; Regeneron, Tarrytown, New York, and Bayer Healthcare Pharmaceuticals, Berlin, Germany) is a 115 KDa recombinant fusion protein that works as a decoy receptor that binds VEGF-A, VEGF-B, and PIGF with a greater affinity than the body’s native receptors. The vitreous half-life of aflibercept in non-vitrectomized animal models is 4 days, less than bevacizumab (7–9 days) and greater than ranibizumab (2.51 days) [28]. Because it has an intact Fc region, it is likely to be subject to FcRn recycling, which is supported by a serum half-life of approximately 5–6 days following intravenous administration [21,39]. Through the 52 weeks of the phase 2 DA VINCI study, patients in the four groups of aflibercept: 0.5 mg aflibercept every 4 weeks (0.5q4), 2 mg aflibercept every 4 weeks (2q4), 2 mg aflibercept for three initial monthly doses and then every 8 weeks, (2q8), 2 mg aflibercept for three initial monthly doses and then on an as-needed (Pro Re Nata-PRN) basis, experienced mean BCVA gains from baseline to week 24 ranging from 8.5 to 11.4 letters compared with only 2.5 letters in the laser PRP group. Because of its improved binding properties, aflibercept represented an opportunity to potentially reduce the treatment burden and follow-up visits to every 8 weeks [22]. VIVID and VISTA, two similarly designed phase 3 trials, showed that after 52 and 100 weeks of treatment aflibercept provides significant and similar sustained improvement with 2q4 and 2q8 dosing regimens. Moreover, the percentage of patients with a ≥2-step improvement in ETDRS-DRSS score from baseline to 1 year was significantly greater in the aflibercept groups compared to laser [23].

Reduction in potentially vision-threatening complications, center-involvement DME or PDR, in eyes with moderate to severe NPDR using four doses of aflibercept (basal, 1, 2, and 4 months) and then every 4 months through 2 years was observed in DRCRNet protocol W. The 2-year cumulative probability of developing center-involvement DME or PDR was 16.3% with aflibercept vs 43.5% with sham; nevertheless, preventive treatment did not confer BCVA benefit compared with observation plus treatment with aflibercept only after the development of PDR or vision-reducing center-involvement DME [40]. These findings support results obtained in the PANORAMA study, in which patients with moderate to severe NPDR without center-involvement DME at two years showed ≥2-step improvement in ETDRS-DRSS: 62% of the eyes that received aflibercept every 8 weeks, and 50% of the eyes that received aflibercept every 16 weeks vs 13% in the sham group. The PANORAMA study demonstrated at week 24 that aflibercept improved the severity of DR and suggests that anti-VEGF can reverse disease progression in these patients [41]. The DRCRNet protocol AB compared aflibercept vs vitrectomy plus PRP for patients with PDR and vitreous hemorrhage; at 24 weeks no significant difference in BCVA was found between groups. Interestingly, over the 2 years, 33% of eyes assigned to aflibercept needed vitrectomy and 32% of patients in the vitrectomy plus PRP received subsequent aflibercept; we need to take these results with caution due to the high range of confidence intervals in the results [42]. Currently, high-dose (8 mg) aflibercept is being studied in the phase 3 trial PHOTON, looking for non-inferiority, efficacy, and safety of high-dose aflibercept applied at 12- or 16-week intervals compared with 2 mg aflibercept every 8 weeks for DME treatment [43].

#### 3.1.4. Brolucizumab

Brolucizumab is a 26 kDa single-chain antibody fragment (scFv) that has a high affinity for VEGF. Its low molecular weight allows for a greater drug concentration per injection and offers the potential for more tissue penetration and increased duration of action [44]. Recently, the 52-week outcomes of the two phase 3 pivotal trials KESTREL and KITE were published. Brolucizumab 6 mg was given in five loading doses every 6 weeks followed by dosing every 12 weeks. It achieved non-inferiority compared with aflibercept 2 mg given in five loading doses with a 4-week interval followed by every 8 weeks at 52 weeks. The probabilities for exclusively maintaining q12 dosing after loading through week 52 were 55.1% for the 6 mg arm in KESTREL and 50.3% in KITE. As expected, an additional benefit was observed in the ETDRS-DRSS score; 28.6%, 29.6%, and 21.7% of the patients experienced a ≥2-step improvement in the brolucizumab 3 mg, 6 mg, and aflibercept 2 mg groups from baseline at 52 weeks, respectively. Reported ocular secondary adverse events, especially intraocular inflammation (including retinal vasculitis), retinal vascular occlusion, and endophthalmitis, were of special interest due to the higher incidence of these adverse events reported elsewhere. However, the incidence of severe intraocular inflammation (IOI) with brolucizumab seems lower in DME than nAMD [25,36,37].

#### 3.1.5. Conbercept

In 2013, conbercept (KH902; Chengdu Kanghong Biotech Co., Chengdu, China) was approved in China for the treatment of nAMD [45]. It is a recombinant fusion protein composed of the second immunoglobulin (Ig) domain of VEGFR1 and the third and fourth Ig domains of VEGFR2 to the Fc of human IgG1 [46]. It has similar properties to aflibercept. The Sailing study was a multicentre, randomized, double-masked parallel controlled, phase 3 trial, whose objective was to compare the efficacy and safety of intravitreal conbercept injections versus laser for the treatment of DME; at 1-year completion of the study an extra 1-year follow-up extension with crossover to conbercept was made. A significant improvement in BCVA from baseline to month 12 was observed in the conbercept group, whereas no improvement was observed in the laser group, with a similar outcome in patients who followed treatment in the extension year. Furthermore, patients in the laser group that crossed over to conbercept PRN demonstrated a significant change from months 12 to 24 [24]. A randomized clinical trial comparing other approved therapies with conbercept is needed to establish non-inferiority.

#### 3.1.6. Faricimab

Faricimab is the first bispecific antibody designed for intraocular use. The angiopoietin (Ang) and tyrosine kinase with immunoglobulin-like and epidermal growth factor homology domains (Tie) signaling pathway is a key regulator of vascular stability, and Ang-2 upregulation has been implicated in the pathogeneses of DME and other retinal vascular diseases [47,48]. Its antigen-binding fragments independently inhibit Ang-2 and VEGF-A with high affinity and specificity, while its fragment crystallizable (Fc) region was engineered to reduce Fc-mediated effector functions and systemic half-life [49]. In the phase 2 BOULEVARD trial, faricimab demonstrated statistically significant superior gains at week 24 in treatment-naïve patients randomly assigned to faricimab 6.0 mg every 4 weeks versus ranibizumab 0.3 mg every 4 weeks [50]. In light of these findings, two phase 3 studies, YOSEMITE and RHINE, further investigated faricimab for DME. Both studies reached their primary efficacy endpoint, each demonstrating non-inferior 1-year vision gains with faricimab every 8 weeks or personalized treatment interval (PTI, a modified treat-and-extend—T&E regimen) versus aflibercept every 8 weeks. Decreased treatment burden was demonstrated in YOSEMITE and RHINE, with more than 70% of patients in the T&E groups achieving every-12-week dosing or longer at 1 year. At the week 52 visit, 151 (53%) patients in YOSEMITE and 157 (51%) patients in RHINE achieved dosing every 16 weeks, and a further 60 (21%) patients in YOSEMITE and 62 (20%) patients in RHINE achieved dosing every 12 weeks. These findings highlight the potential role of faricimab in achieving an unmet goal of durable therapies that optimize real-world outcomes [26].

#### 3.1.7. Other Anti-VEGF Therapies

Another antibody biopolymer conjugate (ABC) KSI-301, with molecular weight 950 kDa (KODIAK sciences inc., Palo Alto, California) is under trial for the treatment of nAMD, macula edema secondary to retinal vein occlusion (RVO) and DME. The ABC includes a humanized IgG1 antibody with inert immune effector function and a biopolymer which is a high molecular weight phosphorylcholine polymer covalently bound by single-site specific linkage; this enhances its size and molar dose to increase intraocular durability [51]. In rabbit models it binds and blocks all isoforms of VEGF-A with higher affinity than its intended receptors VEGFR1 and VEGFR2; ocular half-life, also in rabbit models, has been demonstrated to be more than 10.5 days in the retina and more than 12.5 days in the choroid [52]. GLEAM and GLEAMER are both multicenter, randomized, phase 3 studies, whose main goal will be to evaluate the safety, efficacy, and durability of KSI-301 in the treatment of naïve patients with DME compared to aflibercept. KSI-301 will be administered every 8–24 weeks after three loading doses vs aflibercept every 8 weeks after five loading doses; both studies’ endpoints change in BCVA from baseline at 1 year, and treatment will be followed up for 2 years. Another phase 3 study with aims to evaluate this ABC administered every 4 or 6 months after two bimonthly doses in patients with NPDR at 1 year is the GLOW study, which is intended to recruit close to 400 patients [53].

Adjuvant therapies to existing ones are beginning to be explored. A trap agent, OPT-302 (Opthea Limited), binds and neutralizes VEGF-C and -D. A multicenter phase 1b/2a trial evaluated this molecule in combination with aflibercept for refractory DME. A total of 53% of patients treated with combination therapy gained ≥5 letters at week 12 compared with the baseline, which was greater than the predefined success measure of 38%. In a subgroup of patients with a prior history of aflibercept treatment, after switching to combination therapy the mean change in BCVA at week 12 was +6.6 letters and +3.4 letters for those continuing monotherapy [54].

### 3.2. Steroids

Anti-VEGF agents were the first approved therapy for DME, however not every patient responds to these molecules; as many as 30–40% of patients in clinical trials did not reach a BCVA of 20/40, nor did they gain ≥5 letters [55,56]. Moreover, some patients with a recent history of arteriothrombotic events can be contraindicated to anti-VEGF. For those patients, and even in treatment-naïve ones, alternative treatments such as one or more preparations of corticosteroids may be an option [57,58].

In the pathophysiology of DME, other molecules and pathways are up-regulated besides VEGF such as intracellular adhesion molecule (ICAM)-1, tumor necrosis factor (TNF) α, cyclooxygenase-2, and interleukin-6 (IL-6), neutrophils and monocytes are attracted, and vascular permeability deteriorates [59]. Steroids act by downregulating the arachidonic acid pathway, reducing the synthesis of thromboxane, prostaglandins, and leukotrienes, indirectly reducing VEGF synthesis. Corticosteroids inhibit the inflammatory processes involved in DME, including the production of pro-inflammatory mediators, increased levels of VEGF, and the loss of endothelial tension-binding proteins [60,61,62]. Currently available steroid therapies for the treatment of DR and DME are intravitreal and sub-tenon triamcinolone acetonide (TA), the dexamethasone intravitreal implant, and the fluocinolone intravitreal implant.

#### 3.2.1. Triamcinolone Acetonide

The administration of intravitreal TA has demonstrated short-term efficacy (<3 months) for patients with DME unresponsive to at least three doses of bevacizumab with gains of BCVA at 1 month and sustained gains at 2 months but not at 3 months [63]. Compared to intravitreal, sub-tenon administration is less effective in terms of visual gains and anatomic outcomes with no difference at 6 months; patients with intravitreal treatment had a higher intraocular pressure (IOP) response at 3 months with no difference at 1 month and a lower IOP at 6 months [64]. Due to its short duration and the necessity of multiple injections, monotherapy for DME is not usually employed but is a good alternative in combination with anti-VEGF to achieve faster gains in visual acuity, anatomic outcomes, and, more importantly, decrease the number of anti-VEGF injections in the treatment of DME [65].

#### 3.2.2. Dexamethasone

Dexamethasone is a water-soluble corticosteroid that can be delivered to the vitreous cavity by the dexamethasone intravitreal implant (DEX implant; OZURDEX, Allergan, Inc., Irvine, CA, USA). The implant is composed of a biodegradable copolymer of lactic acid and glycolic acid containing micronized dexamethasone. It releases the total dose of dexamethasone over consecutive months after insertion; its efficacy has been demonstrated with similar anatomic outcomes compared to anti-VEGF therapy but with lower BCVA gains in phakic patients at 12 months, this may be due to the progress of lens opacification with treatment. In addition, patients treated with the implant needed fewer retreatment injections compared to anti-VEGF. Dexamethasone implants may be considered in patients in which anti-VEGF is contraindicated, in pseudophakia eyes, in patients who do not want to be treated frequently, and in those not responsive to anti-VEGF [66,67,68]. In a real-world setting, treatment naïve DME patients treated with DEX-implants demonstrated a gain of ≥15 letters in almost 60% of eyes [69,70]. Concerns of adverse effects such as ocular hypertension and the necessity of filtration surgery for secondary glaucoma may be a bias toward the preference for anti-VEGF treatment, but in a randomized clinical trial the need for glaucoma filtering surgery was 0.3% and no patient needed removal of implants for IOP control [71]. The safety profile has been widely studied in the literature [72,73]. There are clinical trials that have shown some benefits of intravitreal steroids in the progression of DR [74]. The DR-Pro-Dex study provides the first long-term evidence that the dexamethasone implant has the potential to not only delay the progression of DR and PDR but may also improve the severity of DR in 24 months [75]. On the other hand, the results of the TRADITION study conclude that the implantation of dexamethasone at the end of a pars plana vitrectomy (PPV) in patients with tractional diabetic retinal detachment improves the severity of PDR and reduces the detachment rates [76].

A phase 3 trial is underway for a topical formulation of dexamethasone, OCS-01 (Oculis, 1.5% ophthalmic suspension), using the nanoparticle technology for delivery. Phase 2 clinical trials showed a decrease in the mean central macular thickness in the treatment arm compared to control eyes, reaching its primary endpoint; moreover, the treatment arm at week 12 had higher gains in visual acuity in letters compared to the control [77].

Another intravitreal therapy using dexamethasone is AR-1105 (Arie, Pharmaceutics, Inc.), which consists of a bio-erodible intravitreal implant manufactured using PRINT^®^ technology, designed to last at least 6 months with a lower dose of dexamethasone than current therapies. In phase 2 studies, it showed improvements in BCVA and macular edema at 6 months. Phase 3 studies are underway [78].

The suprachoroidal delivery of dexamethasone is also studied currently in the form of biodegradable microspheres, OXU-001 (Oxular, Limited). A preclinical study of this therapy in rabbits found that therapeutic levels were maintained for approximately 1 year. The drug is administered via pars plana with a microcatheter designed to target the suprachoroidal space [79]. The company is developing a phase 2 study of OXU-001 in patients with DME.

#### 3.2.3. Fluocinolone Acetonide Implant

Fluocinolone acetonide intravitreal implant (Iluvien^®^, Alimera Sciences Ltd., Alpharetta, GA, USA), with an average release at 0.25 µg/day and a described duration of 36 months, was approved by the National Institute for Health and Care Excellence (NICE) in 2013 as a treatment option for DME in pseudophakia patients resistant to anti-VEGF therapies. In the FAME studies, only 25% of patients needed more than one injection at 36 months. Real-world results are similar to those reported in the FAME studies, with a substantial gain of vision in patients treated [80,81,82]. Concerning raised IOP with treatment, 25% of patients in the FAME studies needed antihypertensive drops, and 14.3% needed glaucoma surgery despite topical therapy; these results not being supported by real-life studies demonstrated fewer ocular hypertension issues [83,84]. Careful selection of patients is important to avoid complications [85].

#### 3.2.4. Other Steroid Therapies

A Bcl-xl inhibitor is currently under investigation for treating DME or nAMD in patients where anti-VEGF therapy is not considered beneficial. In phase 1 studies, UBX1325 (Unity biotechnology) at 24 weeks demonstrated that 50% of patients had a ≥10 letter gain and 62.5% experienced a ≥5 letter gain. A phase 2a study is currently enrolling patients with DME, aiming to measure ocular and systemic safety and tolerability over 24 weeks [86].

THR-149 (Oxurion) is a plasma kallikrein inhibitor given as an intravitreal injection. At 90 days of the phase 1 trial, 12 patients had an average improvement of 6.4 letters after one dose. A phase 2 trial, KALAHARI, is enrolling 122 patients to randomly assign three monthly injections of THR-149 or aflibercept; the primary outcome will be the mean change in vision from baseline at 3 months. [87]

## 4. Discussion

Success in the treatment of PDR and DME lies in the appropriate regulation of VEGF and pro-inflammatory factors. Pharmacologic treatment seems to achieve this goal more effectively compared to retinal laser therapy for DME. However, based on the results of the multiple studies mentioned, the pharmacologic treatment seems to emerge as first-line therapy in PDR, but can be only used at this date for delaying PRP. Although, an individualized treatment considering patient adherence, economic burden, and transportation is key to successful outcomes. For example, a patient with a low level of education, no family support, whose home is far from your practice, with PDR with or without DME, may have more benefit from initial PRP than anti-VEGF monotherapy; if possible, the PRP could be applied in one sitting.

The goal of extended durability and greater efficacy for the treatment of DME and DR is increasingly close, due to the discovery of new pathways and molecules involved in its pathogenesis. The recently approved faricimab is an example of this, with more than 70% of the patients with DME in the T&E arm achieving an every-12-weeks or more dosing regimen. We need to wait for real-world data to know if these results are reproducible and adjust to our patients’ needs.

In addition to the new molecules in development presented above, many other pathways are involved in DR pathogenesis. This is the case for the AGEs and their receptors (RAGEs). AGEs bind to RAGEs and trigger the production of reactive oxygen species (ROS), with upregulation of the pro-inflammatory molecule NFκ-β [88]. AGEs also increase VEGF expression; this makes it a target for novel treatments to improve microvascular damage in DR. With the objective of blocking or reducing the effects of this pathway, two molecules had been tested in induced diabetic rats: silymarin and fangchinoline [89,90]. The former is a flavonoid compound isolated from *Silybum marianum* with antioxidant and anti-inflammatory properties. In diabetic rats, it demonstrated an improvement through a decrease in AGEs, RAGEs, and ROS levels, a reduction in p38 MAP kinase and NFκ-β pp65 phosphorylation which was reflected in the inhibition of adhesion molecules and extracellular matrix proteins. The latter is an alkaloid isolated from *Stephania tetranda*; it acts by attenuating the expression of IL-6, IL-1, TNF-α, and cyclooxygenase [89,91]. Treatment with fangchinoline reduced the plasma concentration of HbA1c, glucose, and AGEs in the retinal tissues of DR rats [90]. Currently, there are only preclinical studies of these molecules, but their results are promising; they may not be more effective than blocking the well-known VEGF pathway, but may work as an adjuvant treatment to decrease treatment burden or increase the effectiveness of intraocular therapies.

Another area of research is the role of damage-associated molecular patterns (DAMPs) which are endogenous danger molecules released from the extracellular and intracellular space of damaged tissue or dead cells. These molecules are expressed in different eye diseases, and diabetic retinopathy; the predominant DAMPs are S100, HMGB1, uric acid, HSPs, ATP, cyclophilin A, Aβ, IL 1-α, IL-33, nuclear DNA, mtROS, formyl peptide, and lipid from the mitochondrial membrane [92]. DAMPs can act on the RAGE, activating the NFκ-β pathway, and activating the transcription of cytokines, chemokines, and other inflammatory mediators involved in DR. These DAMPs can therefore be targeted by novel therapeutics or repurpose existing molecules. In this way, potential treatment prospects are used in different fields of medicine; one of them is tasquinimod, which is used in prostate cancer, and its mechanism of action is poorly understood. It is thought to have an antiangiogenic effect inhibiting myeloid-derived suppressor cells, and down-regulation of hypoxia-inducible factor-1α [93]. Other prospects are the regulators of sirtuins (SIRTs), an enzyme family that comprises seven isoforms whose main action is the modulation of DNA repair, cell cycle, metabolism, and aging. SIRT 1,3,5, and 6 are regulators of DR; they control sensitivity to insulin, glycolysis, gluconeogenesis, and the initiation of the inflammatory process [94,95]. A glycoside extracted from the roots of the licorice plant, glycyrrhizin, when administered to diabetic animal models was able to reduce ROS, IL-1β, TNF-α, cleaved caspase 3 levels, along with retinal vasculature permeability [95]. These two pathways look promising for future human trials and need to be considered as potential targets for DR and DME treatment.

Treatment decisions may be different for each manifestation and may be modified based on its behavior. We must not forget that both DME and PDR are different manifestations of DR and therefore must be assessed individually. We propose, therefore, a treatment algorithm including these two diabetic complications (Figure 1) [96].

Several protocols are currently being developed to better understand the behavior of PDR and DME in different settings and to provide a stronger foundation for an effective and timely treatment regimen.

## 5. Conclusions

We are facing an exciting era in the development of new, upcoming drugs. Considering the studies that shed light on new molecules and pathways involved in the DR and DME pathogenesis, companies and investigators are placing all their efforts into developing more effective and durable therapies to reach the goals of lower treatment burden and gains in visual acuity for these patients. The control of glycemic, blood pressure, and lipid levels is essential for good results; without proper control, any therapy is destined to fail.

## Figures and Tables

**Figure 1 pharmaceutics-15-00122-f001:**
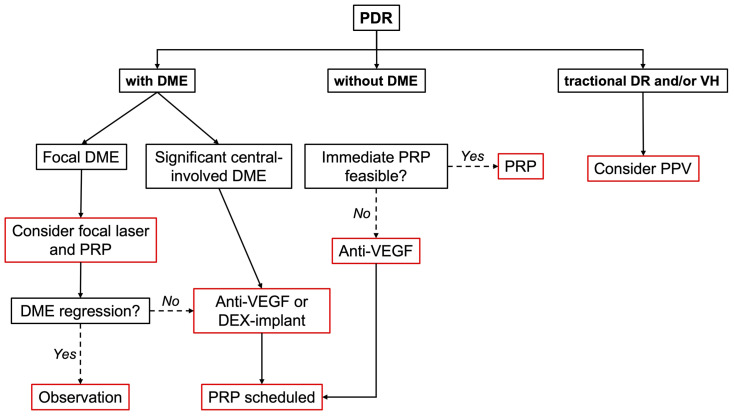
Treatment flow chart in different PDR scenarios. DEX-implant: dexamethasone implant; DME: diabetic macular edema; PDR: proliferative diabetic retinopathy; PPV: pars plana vitrectomy; PRP: panretinal photocoagulation; VEGF: vascular endothelial growth factor; VH: vitreous hemorrhage. If starting anti-VEGF for DME, PRP can be deferred since the same anti-VEGF may control both DME and PDR. Consider factors such as risk of non-compliance, treatment cost, and treatment burden. Cases with TRD should not receive only anti-VEGF therapy due to increased traction progression risk. However, anti-VEGF injection can be applied a few days before PPV is performed to decrease intraoperative and postoperative VH.

**Table 1 pharmaceutics-15-00122-t001:** Anti-VEGF therapies in diabetic macular edema and diabetic retinopathy.

Drug	Mechanism of Action	Dose	Gain in Letters in DME	Change in DR	Studies
Pegaptanib	Pegylated oligoribonucleotide (aptamer), binds to VEGF_165_	0.3 mg q6	1 year: +5.2 2 year: +6.1	Reduction of NV in eight out of 19 patients	[18,19]
Bevacizumab	Humanized murine full-length mAb, binds VEGF-A alone	1.25 mg	1 year (20/32–20/40): +7.5 1 year (20/50–20/320): +11.8 2 year (20/32–20/40): +6.8 2 year (20/50–20/320): +13.3	2 years: 30% improvement *	[20]
Ranibizumab	Humanized murine mAb fragment, binds VEGF-A, higher affinity	0.3 mg 0.5 mg	1 year (20/32–20/40): +8.3 1 year (20/50–20/320): +14.2 2 year (20/32–20/40): +8.6 2 year (20/50–20/320): +16.1	2 years: 38% improvement * Less visual field lost at 5 years vs PRP	[14,20]
Aflibercept	Human fusion protein of the IgG Fc region, binds VEGF-A, VEGF-B, PlGF-1 and PlGF-2	2.0 mg q8	1 year (20/32–20/40): +8.0 1 year (20/50–20/320): +18.9 2 year (20/32–20/40): +7.8 2 year (20/50–20/320): +18.1	2 years: 70% improvement 2 years: 62% improvement 33% need of vitrectomy	[20,21,22,23]
Conbercept	Recombinant fusion protein, Binds VEGF-A, -B, and PlGF	0.5 mg	1 year: +8.6 Laser crossover	No RCT Improvement in NV severity	[24]
Brolucizumab	Single-chain antibody fragment (scFv) with high affinity for VEGF	6 mg q6-q12 dosing 50–55% •	1 year: +9.2 and +10.6	29.6% improvement *	[25]
Faricimab	Bispecific antibody Inhibit Ang-2 and VEGF-A	6 mg >70% achieved q12 in T&E	YOSEMITE 1 year: +10.7 q8 +11.8 PTI RHINE 1 year: +11.8 q8 +10.8 PTI	YOSEMITE 1 year: 46% q8 42.5% PTI RHINE 1 year: 44.2% q8 43.7% PTI	[26]

NV: neovascularization; PTI: personalized treatment interval (YOSEMITE and RHINE); RCT: randomized controlled trial; * Improvement from baseline: ETDRS diabetic retinopathy severity scale level improved by at least two levels or there was complete regression of active proliferative diabetic retinopathy (PDR). • probability of q12 dosing posterior to loading phase through week 52.

## Data Availability

No new data was created; all data is available for public domain.

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
