# Peer review of "Current Treatments and Innovations in Diabetic Retinopathy and Diabetic Macular Edema"

_pharmaceutics, 2022, doi:10.3390/pharmaceutics15010122_

Round 1

Reviewer 1 Report

 The paper has presented treatments in diabetic retinopathy and diabetic macular edema. Although the contribution is minor but significant.

I have the following major corrections to consider in the next phase of evaluation.

1. Please discuss the problem statement in detail and added in detail in the Introduction section. 

2. Add manuscript detail in last of the introduction section, we cannot find any text which describes the manuscript formulation and structure.

3. Mention all dimension in conclusion, no single dimensions are mentioned in conclusion section example total feature vector, selected vector etc. 3. A huge experimentation is performed in this article but where are limitations of the experimentation.

4. Explain the methodology in detail; preferably add pseudo code.

5. Explain the dataset and compare results on the same dataset to the reported state of art

6. What is the research question in this article not mentioned clearly and many grammatical mistakes in related work? 

Reviewer 2 Report

1. The accumulation of Advanced glycation end products (AGEs) in retinal cells is strongly associated with the development of DR. The interaction between AGEs and the receptor for AGE (RAGE) is involved in multiple cellular pathological alterations in the retina.  The capillary endothelial cell degeneration is mediated by activating the AGE-RAGE pathway and increasing MMP activity in endothelial cells by impairing pericyte function in the retina. Interestingly, two Chinese patent medicines were found to include components effective at treating PDR by affecting the pathways such as AGE-RAGEEvid Based Complement Alternat Med. 2021 Mar 5;2021:6642600. . Thus, the RAGEs antagonist FPS-ZM1 (ACS Chem Neurosci. 2021 Jan 6;12(1):63-78.), or the matrix metalloproteinase (MMP) inhibitor GM6001, which may significantly attenuate AGEs-induced capillary endothelial cell degeneration, may be potential therapeutics. Considering the highly pathogenic roles of AGE-RAGE in the diabetic retinopathy, Will the author give some speculation about this point? For example, antioxidant phytochemicals, as AGE formation inhibitors, are a class of chemicals with reductive and biological activities and possess therapeutic potential in DR treatment.

 2. Besides AGE-RAGE, Damage-associated molecular patterns (DAMPs) such as S100 proteins and HMGB1 act on the RAGE receptor. The interactions of DAMPs and the RAGE signaling pathway have also been implicated in proliferative diabetic retinopathy (PDR) and diabetic macular oedema (DME). The interaction of these DAMPs with RAGE receptors activates NF-κB, actuating the transcription of cytokines, chemokines and other inflammatory mediators (CCL2, CCL5, CXCL10, CXCL12 TNF-α, IL-1β, IL6, ICAM-1, VCAM-1, NOS-2) involved in retinal disorders. The excessive production of DAMPs in response to infection or inflammation has led to the discovery of several proteins and molecules that can be targeted to develop novel therapeutics or repurpose existing drugs to treat retinal disorders such as Tasquinimod or GlycyrrhizinInt J Mol Sci. 2022 Feb 26;23(5):2591.. Will the author include some comments about these therapeutics in the manuscript?

Round 2

Reviewer 1 Report

Authors have addressed my comments successfully so accepted as it is